# Wnt5a Regulates Focal Adhesion Formation to Promote Migration in Ewing Sarcoma

**DOI:** 10.3390/cancers17223712

**Published:** 2025-11-20

**Authors:** Alissa Baker, Anusha Singhal, Sarah Jacobson, David M. Loeb

**Affiliations:** 1Department of Pediatrics, Albert Einstein College of Medicine, Bronx, NY 10461, USA; alissa.baker@einsteinmed.edu (A.B.); anusha.singhal@einsteinmed.edu (A.S.);; 2Department of Developmental and Molecular Biology, Albert Einstein College of Medicine, Bronx, NY 10461, USA; 3Montefiore Einstein Comprehensive Cancer Center, Albert Einstein College of Medicine, Bronx, NY 10461, USA; 4Cancer Dormancy Institute, Albert Einstein College of Medicine, Bronx, NY 10461, USA; 5Marilyn and Stanley M. Katz Institute for Immunotherapy for Cancer and Inflammatory Disorders, Albert Einstein College of Medicine, Bronx, NY 10461, USA

**Keywords:** Ewing sarcoma, non-canonical Wnt signaling, metastasis, cell adhesion molecules

## Abstract

Ewing sarcoma is the second most common bone tumor in children, adolescents and young adults. Over the last three decades, the five-year survival rate for metastatic Ewing sarcoma has not improved, highlighting the need to better understand the mechanisms driving metastasis. Wnt signaling has been implicated in driving migration, invasion, and metastasis in various cancers. In this study, we show that a non-canonical Wnt signaling pathway drives Ewing sarcoma migration, with Wnt5a being a key regulator of this mechanism. We demonstrate changes in the cytoskeleton and in cell adhesion molecule expression with the activation of this pathway, supporting the role of non-canonical Wnt signaling in altering cell intrinsic properties to drive migration. Our work highlights the need to target Wnt5a-responsive non-canonical pathways, along with conventional beta-catenin-dependent canonical pathways, when developing clinical trials to improve patient outcomes by preventing metastatic relapse.

## 1. Introduction

Ewing sarcoma (ES) is the second most common malignant bone tumor in adolescents and young adults [1]. Dose-intensified chemotherapy has improved survival for patients with localized disease to greater than 70% today [2]; however, a third of patients who initially present with localized disease will suffer a metastatic relapse. Those patients, as well as patients who initially present with metastatic disease, are faced with an approximate 25% overall survival rate, which has not changed since the introduction of chemotherapy in the 1970s. The most significant prognostic factor in Ewing sarcoma is the presence or absence of metastatic disease [3]. As Ewing sarcoma is a translocation-driven sarcoma, its genetic landscape is otherwise relatively silent with few, if any, targetable mutations [4,5]. Many targeted therapies are currently being studied, but developing targeted therapies focused on and measured by local tumor response fails to address the most pressing need for Ewing sarcoma patients: the prevention and treatment of metastatic disease. Therefore, a better understanding of the biology of Ewing sarcoma metastasis is needed to improve survival rates [6].

Wnt signaling plays an important role in cancer biology, including Ewing sarcoma [7,8,9]. Several genes encoding proteins involved in Wnt signaling are transcriptional targets of EWS-Fli1, the defining translocation of Ewing sarcoma. There is also evidence that Ewing sarcoma cells can modulate the extracellular matrix in a way that potentiates Wnt signaling [10]. One of the challenges of targeting Wnt signaling in any cancer is the complexity of this pathway, which consists of 19 individual ligands that can bind over a dozen distinct receptors activating multiple independent intracellular cascades [11,12]. The best-described Wnt signaling cascade is the canonical beta-catenin-dependent pathway wherein an extracellular signal causes beta-catenin to translocate into the nucleus and activate transcription of a host of target genes. There are numerous non-canonical beta-catenin-independent pathways engaged by Wnt ligands, including the planar cell polarity pathway, which is necessary for embryonic development [13]. The multitude of ligands and receptors has complicated the development of drugs targeting Wnt signaling, but one such compound currently in clinical trials, WNT974, is a Porcupine (Porcn) inhibitor that inhibits all functional Wnt signaling by preventing palmitoylation of all Wnt ligands [14]. Porcn is the only enzyme that mediates Wnt ligand palmitoylation, a necessary post-translational modification that allows the secreted Wnt ligand to tightly bind its receptor and initiate downstream signaling. Wnts may be secreted without the palmitate tail but in the unpalmitoylated state, they are unable to bind to their receptor; therefore, a Porcn inhibitor is a pan-Wnt inhibitor [15].

Our lab has previously reported that WNT974 prolongs metastasis-specific survival in a mouse model of spontaneous Ewing sarcoma metastasis by delaying the development of pulmonary metastases [16]. WNT974 impaired the development of Ewing sarcoma metastases from orthotopically implanted Ewing sarcoma tumors but not from those cells injected by tail vein; therefore, Wnt signaling must regulate an early step in the metastatic process. Cell migration is coordinated through highly dynamic events that rapidly establish and remodel cell–matrix interactions required to generate forces to propel cells. Focal adhesions are large macromolecular structures assembled from intracellular actin bundles linked to the extracellular matrix that generate the mechanical force to deform the membrane, giving the cell directionality and migratory capacity [17,18]. Proteins that localize to focal adhesions can be divided into catalytic proteins, such as the tyrosine kinases FAK and Src, and adaptor proteins, such as ALCAM and MCAM [19,20]. Cell adhesion molecule MCAM has been shown to play a role in Ewing sarcoma migration and metastasis, while ALCAM is expressed in more than 70% of pediatric sarcomas, leading us to focus on cell adhesion membrane family members [19]. Overactivation of FAK has been shown to promote tumorigenesis in numerous cancers, including colon, prostate, breast, and ovary [21]. Src is known to regulate actin dynamics, including directly associating with actin in colon cancer [22]. FAK has been previously identified as a candidate therapeutic target in Ewing sarcoma, supporting a role in biomechanics of the cancer cells [23,24]. Thus, in further evaluation of the early steps of metastasis in Ewing sarcoma, we have identified a non-canonical beta-catenin-independent Wnt signaling pathway, dependent on autocrine/paracrine Wnt5a and involving FAK, Src, and ALCAM, that drives migration of Ewing sarcoma cells through formation of focal adhesions.

## 2. Methods and Materials

### 2.1. Cell Lines and Xenografts

All ES cell lines were obtained through ATCC. Cell line identity was confirmed by STR testing in the Albert Einstein College of Medicine Genomics Core. Cells were cultured at 50–70% confluence in RPMI-1640 medium supplemented with 10% fetal bovine serum (FBS; Invitrogen, Carlsbad, CA, USA) and were routinely confirmed to be Mycoplasma negative using the MycoAlert Plus Mycoplasma detection kit (Lonza, Basel, Switzerland). Using A4573, we generated Wnt5a Crispr-Cas9 clones using Origene (Rockville, MD, USA) Wnt5a Human Gene Knock-out kit with a GFP-puromycin functional cassette (KN209206). Co-transfection with 1 μg of each gRNA or the scramble control with the GFP-puromycin vector were performed using lipofectamine and Opti-Mem media. Following 15 min incubation, the mixture was added drop-wise to 50% confluent A4573 cells that had been plated 24 h prior. The cells were split twice weekly for 3 weeks and subsequently selected via puromycin. Cells were sorted for the GFP-tag with flow cytometry and verified with Western blotting using Wnt5a (Novus, #NBP2-24752, 1:1000, Uppsala, Sweden). Due to a low transfection rate, we then performed monoclonal expansion where the cells were cultured at 50–70% confluence in RMPI-1640 supplemented with 8 μg/mL puromycin and 20% FBS. Two resultant clones, A9.5 and F10.6, demonstrated consistent loss of Wnt5a with bright GFP signal.

A4573: EWS;FLi1 (11;22), type 3 fusionTC71: EWS;FLi1 (11;22), type 1 fusionSKES1: EWS;FLi1 (11;22), type 2 fusionCHLA9: EWS;FLi1 (11;22), type 1 fusionCHLA10: EWS;FLi1 (11;22), type 1 fusionCHLA25: EWS;ERG (11:22), EWSR1 exon 7 with ERG exon 6CHLA32: EWS;FLi1 (11;22), type 1 fusion

All Ewing sarcoma cell lines were obtained in collaboration with Dr. David Loeb’s laboratory.

### 2.2. Migration Assays

All ES cells, including Wnt5a CRISPR clones, were plated in RPMI-1640 media with no FBS at a density of 1 × 10^5^ cells/200 µL/well onto an 8.0 µm pore PET membrane insert (Corning Inc., Somerville, MA, USA). Added to the cells in the insert were either 1 µM WNT974 or DMSO 1:1000. The insert was placed into a 24-well plate containing 500 µL of RPMI-1640 media with 10% FBS supplemented. Cells were allowed to migrate through the membrane over 48 h, the remaining cells in the top chamber were removed with a cotton tipped applicator, and membranes were then fixed in 4% paraformaldehyde. The fixed cells on the bottom chamber were stained in Richard-Allan Hematoxylin2 (Thermo Scientific, Waltham, MA, USA) for 3 min, then counted.

The migration assays using the FAK inhibitor, GSK2556098, and the Src inhibitor, saracatinib, were performed with the cells in RPMI-1640 supplemented with 10% FBS with the drug at the reported concentration in the insert. The Boyden chamber well contained RPMI-1640 supplemented with 10% FBS as chemoattractant.

The migration assays using recombinant Wnt5a were performed with RPMI-1640 supplemented with 10% FBS in both the insert and the well. Recombinant Wnt5a, whether in the insert or well, is the sole chemoattractant in these assays. All migration assays were performed in triplicate.

### 2.3. Protein Extraction and Western Blot Analysis

Total cell extracts were prepared using RIPA buffer with protease inhibitors (sc-24984A, SantaCruz, Santa Cruz, CA, USA) with additional PMSF (1 mM), sodium orthovanadate (1 mM), and sodium fluoride (50 mM) added just before use. Protein concentration was quantified using the Pierce BCA colorimetric assay (#23225, ThermoFisher, Rockville, IL, USA) and quantified against BSA standards with a Multiskan plate imager at 750 nM. Extracted proteins were resolved on 4–15% TGX gels (#4561086, Bio-Rad, Berkeley, CA, USA) and transferred to PVDF membranes using BioRad dry transfer system (BioRad). Immunoblots were developed with indicated primary antibodies and corresponding HRP-tagged secondary antibodies. Blots were scanned using the ThermoFisher myECL imager. Antibodies used included pFAK Tyr397 (Cell Signaling, Boston, MA, USA, #3238, 1:250), pSrc Tyr416 (Cell Signaling, #2101, 1:500), FAK (Cell Signaling, #3285, 1:1000), Src (Cell Signaling, #2108, 1:0000), vinculin (Abcam, #129002, 1:1000), ALCAM (Novus, #NBP2-37358, 1:1000), Wnt5a (Novus, #NBP2-24752, 1:1000), beta-2 microglobulin (Cell Signaling, #59035, 1:1000), and anti-rabbit-HRP and anti-mouse-HRP (Vector labs, Newark, CA, USA, MP7451 and MP7802-15, 1:10,000).

### 2.4. Phalloidin Assays

ES cell lines and Wnt5a CRISPR clones were plated in RPMI-1640 media with 10% FBS in a density of 2 × 10^4^–2 × 10^5^ cells/mL/well into chambers of a Lab-Tek II chamber slide system (NUNC, Rochester, NY, USA) and were treated with 1 μM WNT974 or DMSO 1:1000 for 48 h. Slides were fixed and stained with Cytopainter F-Actin Staining Kit 647 (Abcam, Cambridge, UK) per manufacturer’s instructions, and long cytoplasmic extension changes were quantified. Slides were visualized using a Nikon Spinning Disc confocal microscope and analyzed using ImageJ version 1.54p and Volocity 7 software.

### 2.5. Reverse Transcription-Polymerase Chain Reaction (RT-PCR) and qPCR

RNA was extracted from cultured cells using the RNeasy Mini Kit according to manufacturer’s instructions (QIAGEN Inc., Valencia, CA, USA) and reverse transcribed (Iscript Reverse Transcriptase, Bio-Rad, Hercules, CA, USA). For quantitative PCR, including PCR arrays, 1 µL cDNA was mixed with 10 µL SSO Advanced Universal SYBR Green SuperMix (#1708840, Bio-Rad, Berkeley, CA, USA) and 8 µL ddH_2_O per reaction with commercially available qPCR primers (Bio-Rad) or preloaded 96-well real-time PCR array plates (Bio-Rad). Primers used include B2M, Wnt5a, and ALCAM, all from Bio-Rad. Quantitative real-time PCR were performed using a standard two-step amplification/melt protocol per manufacturer’s instructions. Quantification of gene expression was performed by the ΔΔCt method, normalized to the housekeeping gene B2M, and quality controls including the Reverse Transcription Control assay (Bio-Rad), DNA Contamination Control assay (Bio-Rad), and RNA Quality assay (Bio-Rad) were all performed per manufacturer’s instructions.

### 2.6. Luciferase Reporter Assay

Cells were seeded in a 24-well plate in triplicates, and were transfected using Lipofectamine 2000 (#11668027, Invitrogen, Life Technology, Carlsbad, CA, USA) with either M50 Super 8× TOPFlash (Addgene plasmid #12456, Watertown, MA, USA) or M51 Super 8× FOPFlash (Addgene plasmid #12457) reporter construct (kindly provided by Dr. Randal Moon) and pRLSV40 (Promega, Madison, WI, USA). After 24 h of transfection, cells were lysed and analyzed for expression of firefly and Renilla luciferase activity using the Dual-Luciferase Reporter Assay System (Promega) according to manufacturer’s instructions. The TOPFlash/FOPFlash ratio was calculated following normalization by the Renilla luciferase activity from pRLSV40.

### 2.7. Histopathological Analyses

Formalin-fixed, paraffin-embedded tissue was used for analyses. Sections were cut to 5 mm, deparaffinized, and stained with haematoxylin and eosin (#1.15973, Sigma-Aldrich, St. Louis, MS, USA) or processed for immunofluorescence. Samples were deparaffinized in xylene, rehydrated in graded alcohol, and rinsed in PBS, followed by antigen retrieval with boiling citrate buffer, pH 6. Slides were blocked in TBST (1× TBS and 0.05% Tween 20) with 10% goat serum. Anti-human Wnt5a antibody (Novus, #NBP2-24752, 1:200) was diluted in TBST and 1% BSA, and slides were incubated overnight in a humidified chamber at 4 °C. Slides were washed in TBST prior to application of secondary antibodies of HRP-linked anti-mouse antibodies (anti-mouse-HRP, Vector labs, MP7802-15, 1:10,000) in the dark for 1 h at room temperature in a humidified chamber. Slides were then washed in TBST and the coverslips mounted with ProLong Gold antifade reagent with DAPI (Life Technologies, Carlsbad, CA, USA) for nuclear counterstaining. Haematoxylin- and eosin-stained slides were imaged using a Nikon E600 microscope and photographed with a Nikon CCD digital camera using Elements AR software (Nikon, version 6.10.01).

### 2.8. Immunofluorescence

Cells were grown on fibronectin-coated chamber slides (Thermo Fisher Scientific, Waltham, MA, USA), fixed in 4% paraformaldehyde, and permeabilized with 0.1% Triton X-100. After rinsing in phosphate-buffered saline (PBS), the slides were incubated for 1 h in blocking solution (1% BSA and 5% goat serum in PBS). Blocked slides were incubated with primary antibodies including ALCAM (Novus, #NBP2-37358, 1:1000), vinculin (Abcam, #ab129002, 1:1000), and phosphorylated Fak (Tyr397) (Abcam, #ab81298, 1:200) overnight at 4 °C, washed in PBS, incubated with Alexa Fluor 488-conjugated, donkey anti-mouse secondary antibody (1:1000; Life Technologies), Alexa Fluor 555-conjugated goat anti-rabbit secondary, or Alexa Fluor 647-conjugated goat anti-mouse, plus Dapi 1:1000 for 1 h, and then visualized with a Nikon Spinning Disc confocal microscope, including 2 μM Z-stacks. All image processing was conducted in ImageJ and Volocity software.

### 2.9. Immunoprecipitation

All protein extractions were performed as above. A total of 500 μg of protein was used for immunoprecipitation as followed per Abcam #ab206996 immunoprecipitation kit. 40 μL per reaction of Sepharose beads were used. Binding antibody was ALCAM (Invitrogen, MA5-29859, 1:300) and product was eluted using denaturing conditions with SDS buffer. Following elution and denaturing at 95 °C for 10 min, extracted proteins were resolved on 4–15% TGX gels (Bio-Rad) and transferred to PVDF membranes using BioRad dry transfer system (BioRad). Immunoblots were developed with vinculin (Abcam, ab129002, 1:1000) and ALCAM (Invitrogen, MA5-29859, 1:300) primary antibodies and corresponding HRP-tagged secondary antibodies.

### 2.10. PNGase Assay

PNGase F protocol was performed as per New England Biolabs instructions (#0708S), using 20 μg of glycoprotein in both denaturing and non-denaturing conditions. The extent of deglycosylation was determined by mobility shifts on SDS-PAGE gel.

### 2.11. Invasion Assay

Invasion potential of Ewing sarcoma cell lines was performed as per MilliporeSigma Chemicon QCM Gelatin Invadopodia Assay, Green (ECM670). Following poly-L-lysine and fluorescein-gelatin coating of individual wells of chamber slides (ThermoFisher, 177399), approximately 25,000 cells in 500 μL of RPMI/10% FBS were added to each well, followed by either DMSO or WNT974. The slides were incubated for 48 h and then imaged for areas of black punctae in the background of the green fluorescence, signaling areas of degradation, and subsequently quantitated on Image J/Fiji.

### 2.12. Murine Xenografts

NOD/SCID/IL-2Rγ-null (NSG) mice bred by Johns Hopkins University Research Animal Resources were implanted with 3 mm fragments of either a TC71 cell line xenograft or of the patient-derived xenograft (PDX) EWS1 or EWS4 in the pre-tibial space as previously described [16]. Experiments were conducted with cohorts of either 5 or 10 mice per arm, and mice were randomly assigned to treatment or control arms. WNT974 treatment was administered 5 mg/kg/dose twice daily by gavage, 10 times/week. Animals were euthanized when they exhibited signs and symptoms of pain and suffering, such as hunched posture, reluctance to move, weight loss > 20%, or a “body condition score” of 2/5 or worse. All experiments were approved by the Johns Hopkins University Institutional Animal Care and Use Committee.

### 2.13. Statistical Analysis

Unless otherwise indicated, data from all experiments are expressed as mean  ±  SEM from a minimum of three independent experiments. The data was analyzed using GraphPad Prism software, version 10.6.1, by one-way ANOVA and a *p* value of <0.05 was considered significant. To assess pFAK puncta, ImageJ Find Objects was set to pixel size 50–500. All mean fluorescent intensity data was determined from confocal Z stacks which were analyzed in Volocity.

## 3. Results

### 3.1. Porcn Inhibition Results in an Increase in Endogenous Wnt5a Transcription

We have previously demonstrated that Porcn inhibition significantly delays time to development of metastases in Ewing sarcoma xenografts. To further define the signaling pathway that drives this phenotype, numerous Wnt ligands and downstream signaling molecules were evaluated by real-time PCR across seven Ewing sarcoma cell lines after Porcn inhibition (Appendix A). We noted the striking pattern of increased transcription of Wnt5a, a non-canonical Wnt ligand, upon Porcn inhibition in most of the cell lines (Figure 1A). This was the only consistent pattern noted across all ligands we evaluated. Of note, CHLA9 and CHLA10 are derived from the same patient; however, CHLA9 was derived from the primary localized thoracic tumor at the time of diagnosis whereas CHLA10 was derived from a pleural-based metastasis at the time of relapse. CHLA10, the metastasis-derived cell line, demonstrates a statistically significant increase in Wnt5a transcription upon Porcn inhibition while its localized counterpart, CHLA9, does not.

Next, we evaluated whether this same pattern could be appreciated in Ewing sarcoma xenografts that had been treated with Porcn inhibitor. As shown in Figure 1B, Ewing sarcoma xenografts in mice treated with WNT974 also demonstrated a statistically significant increase in endogenous Wnt5a transcription. In addition, the increased Wnt5a transcription correlates with markedly increased expression of Wnt5a protein in the Ewing sarcoma xenografts (Figure 1C), demonstrating that tumors in vivo respond to WNT974 similarly to cell lines in vitro.

To ensure that the Porcn inhibitor, WNT974, was inhibiting palmitoylation of Wnt5a, we performed an acyl–biotin exchange assay (ABE) which confirmed this on target effect. The acyl–biotin exchange technique allows for evaluation of sites of S-palmitoylation based on the labile nature of a thioester bond between palmitate and cysteine residues. Hydroxylamine can cleave that labile bond, resulting in an exposed sulfhydryl group that can subsequently be labeled with biotin and pulled down using streptavidin. Using Tris-HCl as a control for comparison to hydroxylamine treatment allows for direct comparison of the state of palmitoylation of the protein of interest, in this case, Wnt5a. This assay demonstrates that there is less palmitoylated Wnt5a present in the Ewing sarcoma cells treated with WNT974, reflected by the fainter band in WNT974-treated samples compared to control (Appendix A).

### 3.2. Ewing Sarcoma Cells Are Responsive to Endogenous and Exogenous Wnt5a in a Feedback-Dependent Manner

Wnt5a is typically considered to be a non-canonical Wnt ligand, meaning that it mediates signaling through a beta-catenin-independent mechanism. To confirm that there was no significant activation of the canonical, beta-catenin-dependent signaling pathway by recombinant Wnt5a in Ewing sarcoma cell lines, we performed a TOPFlash assay for beta-catenin-dependent transcriptional activity (Figure 1D). In this assay, cells are transduced with a firefly luciferase gene under the transcriptional control of an artificial beta-catenin-responsive promoter. Therefore, Luciferase activity in transduced cells reflects activation of beta-catenin. Recombinant Wnt5a does not activate beta-catenin-dependent signaling to a significant degree in Ewing sarcoma cells. In contrast, Wnt3a, a canonical activator of this pathway, induces significant luciferase activity. Therefore, any transcriptional, translational, or migratory changes seen in the Ewing sarcoma cells treated with exogenous recombinant Wnt5a result from activation of a beta-catenin-independent Wnt signaling pathway.

As WNT974 treatment increased transcription (Figure 1A,B) and protein abundance (Figure 1C) of Wnt5a, we posited that these increases might represent an attempt to compensate for the loss of functional Wnt5a. To test this hypothesis, we treated Ewing sarcoma cell lines with WNT974 to inhibit production of all endogenous Wnt ligands and then added exogenous recombinant Wnt5a to determine if the transcriptional changes are a direct result of Wnt5a in the microenvironment. Strikingly, we found a dose-dependent suppression of Wnt5a transcription in A4573 and CHLA10, two of the cell lines that demonstrated an increase in endogenous Wnt5a transcription upon Porcn inhibition (Figure 1E). The highest concentration of recombinant Wnt5a inhibited the transcription of endogenous Wnt5a to below baseline transcription, and with decreasing amounts of exogenous recombinant Wnt5a, there is an increase in endogenous Wnt5a transcription, reflecting reduced suppression. These results suggest that the cells not only sense and respond to Wnt5a in their environment, but that they also modulate Wnt5a transcription depending on the amount in the environment. Supporting the potential for this feedback regulation of Wnt5a to play a role in the development of metastases, we observed this phenomenon in the metastatic cell line, CHLA10, but not in the cognate cell line, CHLA9, derived from the patient’s primary, localized tumor (Figure 1E). Both cell lines demonstrate a suppression of endogenous Wnt5a transcription by exogenous Wnt5a, but the metastatic cell line, CHLA10, exhibits increased transcription after WNT974 treatment and less suppression of endogenous Wnt5a with lower doses of exogenous ligand, whereas the cognate, nonmetastatic cell line, CHLA9, shows profound suppression of endogenous Wnt5a regardless of the amount of exogenous ligand.

### 3.3. Porcn Inhibition Decreases Ewing Sarcoma Cell Migration in Cells That Are Wnt5a-Responsive

Cell migration is a critical early step in the metastatic cascade. To evaluate the effects of Porcn inhibition on Ewing sarcoma cell migration, cells were treated with either vehicle control or WNT974, and migratory activity was quantified using Boyden chamber assays. With FBS as the chemoattractant, A4573, TC71, and CHLA10 ES cell lines demonstrated a statistically significant decrease in migration when WNT974 was included in the upper chamber, while SKES-1 and the localized ES cell line CHLA9 were insensitive to WNT974 treatment (Figure 2A). These results are consistent with the differential Wnt5a transcriptional effects of these cells after treatment with a Porcn inhibitor (Figure 1A), where we found that SKES-1 and CHLA9 were the only cell lines not to respond to WNT974 by upregulating Wnt5a mRNA expression.

Next, we tested the impact of recombinant Wnt ligands on ES cell migration. We performed Boyden chamber migration assays with recombinant Wnt3a or Wnt5a. Migration of A4573 cells toward FBS was inhibited by WNT974 in the upper chamber. When used as a chemoattractant in the lower chamber, Wnt5a strongly stimulated the migration of A4573 cells, an effect not seen in response to an equimolar amount of Wnt3a (Figure 2B), demonstrating the importance of non-canonical, rather than beta catenin-dependent, Wnt signaling for this process. Because our results demonstrated a feedback-loop regulating Wnt5a transcription, we next evaluated if this feedback pathway can regulate other cellular activities. We measured cell migration in Boyden chamber assays upon dual WNT974 treatment plus rWnt5a either in the upper chamber with the cells or in the lower chamber as the chemoattractant. When rWnt5a was present with the cells in the upper chamber, there was minimal migration through the pores, as compared to vehicle control, whereas we observed a dose-dependent increase in migration when Wnt5a was the chemoattractant in the lower chambers. This migration was most significant at the lowest dose of exogenous Wnt5a. Notably, when WNT974 is present in the upper chamber with rWnt5a in the lower chamber as a chemoattractant, cell migration is completely ablated (Figure 2C). These data are consistent with Ewing sarcoma cells being attracted to the lowest exogenous rWnt5a concentration, consistent with auto-regulatory feedback, and this migration being inhibited by WNT974.

### 3.4. Wnt5a Affects the Structure and Function of Focal Adhesions

As WNT974 and rWnt5a altered ES cell migration, we posited the mechanism could lie in the regulation of focal adhesions. We started our investigation of the impact of Wnt signaling on focal adhesions with the Src family of tyrosine kinases which are known to play a significant role in cell migration, including in Ewing sarcoma [25,26,27]. Western blot analysis demonstrated a decrease in phosphorylation of FAK at tyrosine 397 and, to a lesser extent, Src at tyrosine 416 upon treatment with WNT974, suggesting that both tyrosine kinases are affected by Porcn inhibition (Figure 3A). Tyrosine 397 is the primary autophosphorylation site on FAK and phosphorylation of this residue occurs when FAK binds to the intracellular component of a cell surface receptor. Once that binding occurs, FAK will dimerize, allowing for recruitment and activation of Src [28]. FAK is typically located at sites of focal adhesions where cell membrane protrusions are converted into invadopodia [29,30]. Therefore, we performed immunofluorescence to evaluate whether there was a change in distribution of pFAK (Tyr 397) in Ewing sarcoma cells with WNT974 treatment. As shown in Figure 3B, pFAK can be found in the protrusions of vehicle control-treated A4573 cells whereas it is decreased in number and spread within the cell upon WNT974 treatment.

We next evaluated the functional role of FAK and Src phosphorylation in Ewing sarcoma migration. As evaluated in Boyden chamber assays, a commercially available FAK inhibitor, GSK2256098, inhibited cell migration to an extent similar to that of treatment with WNT974 (Figure 3C). Saracatinib, a commercially available Src inhibitor, also inhibited cell migration to a similar extent (Figure 3D). Taken together, our results support the model that Wnt5a-dependent signaling drives phosphorylation of FAK and Src, and that both FAK and Src activation are necessary for Wnt5a-dependent Ewing sarcoma cell migration.

Due to the significant changes in cell structure and redistribution of ALCAM upon treatment with WNT974, we next evaluated potential adaptor proteins that could link filamentous actin bundles to ALCAM when forming cellular protrusions. Vinculin is a cytoskeletal protein that is involved in cell-to-matrix junctions where it functions to anchor F-actin to the membrane [35]. Using immunofluorescence, we evaluated the presence and distribution of vinculin in Ewing sarcoma cells treated with WNT974 (Figure 4A, Appendix A). Notably, vinculin is present throughout the cells, like ALCAM, with a significant portion in the well-formed protrusions in vehicle control-treated cells. In contrast, vinculin becomes centralized in location, similar to ALCAM, upon treatment with WNT974. To determine whether this protein interaction is affected by Wnt5a, we first treated the cells with recombinant Wnt5a alone or in combination with WNT974 and stained for ALCAM and vinculin. Cells treated with recombinant Wnt5a demonstrated a decrease in both ALCAM and vinculin (Figure 4A), supporting the hypothesis that the interaction of these two proteins is affected by Wnt5a in the tumor microenvironment. In contrast, when the cells are treated with WNT974 to block all functional endogenous Wnt signaling, and then given recombinant Wnt5a, ALCAM and vinculin appear to interact more, similar to the vehicle control cells (Figure 4B). In addition, phalloidin staining of filamentous actin is decreased per cell with WNT974 treatment (Figure 4C). These results suggest that inhibition of non-canonical Wnt signaling leads to a dramatic reorganization of both the actin cytoskeleton and cell adhesion molecules in Ewing sarcoma cells.

By co-immunofluorescence assessment, there appears to be substantial overlap of both ALCAM and vinculin, but this type of analysis does not directly demonstrate a physical interaction between proteins. To confirm that vinculin and ALCAM physically interact, we performed co-immunoprecipitation experiments. These demonstrated a physical interaction between vinculin and ALCAM in vehicle control-treated cells that was increased upon treatment with WNT974. Treatment with the FAK inhibitor, GSK2556098, completely dissociated the two proteins, demonstrating that catalytically active FAK is necessary for this interaction, and suggesting that either too much contact or no contact is detrimental to cell migration, which would be consistent with the need for turnover of focal adhesions for forward migration (Figure 4D).

Because no changes in ALCAM protein levels were appreciated in ES cells treated with WNT974, we postulated that there might be a difference in post-translational modifications. ALCAM function is regulated, at least in part, by palmitoylation. Acyl protein thioesterases 1 and 2 (APT1 and APT2) are cytoplasmic depalmitoylases that act on membrane proteins, such as ALCAM [36]. To determine if ALCAM palmitoylation plays a role in Ewing sarcoma migration, we treated our Ewing sarcoma cells with palmostatin B, a small molecule inhibitor of APT1, and evaluated the cells’ ability to migrate. As shown in Figure 3D, palmostatin B treatment led to a statistically significant decrease in cell migration in CHLA10, but not CHLA9, in Boyden chamber assays. In addition, when A4573 cells were treated with palmostatin B, immunofluorescence analysis revealed that ALCAM became more diffuse and punctate throughout the cells with a visible decrease in vinculin, suggesting that these proteins all interact in Ewing sarcoma cells (Figure 3D).

### 3.5. Wnt5a Knock-Out Ewing Sarcoma Cells Phenocopy WNT974 Treatment

Our data suggests that Wnt5a is a major regulator of Ewing sarcoma cell migration, and that inhibition of a Wnt5a-dependent autocrine/paracrine signaling pathway is the mechanism by which WNT974 suppresses Ewing sarcoma metastasis. To prove this point, we generated Wnt5a knock-out Ewing sarcoma cell lines. Using two different gRNAs, through puromycin selection and monoclonal expansion, we generated two independent A4573-derived cell lines with Wnt5a knocked out via CRISPR/Cas9 gene editing, designated A9.5 and F10.6. On Western blotting, both clones demonstrate a lack of Wnt5a (Figure 5A). In culture, Wnt5a knock-out clones display a distinct growth pattern. In contrast to parental A4573 cells and scramble controls, which grow in a monolayer of spindle cells, the clones, A9.5 and F10.6, grow as clumps, more adherent to each other than to the tissue culture dish, and only adhere well to the dish with a coat of fibronectin (Figure 5B). In addition to their altered growth pattern, the Wnt5a knock-out clones demonstrate a significant decrease in migratory activity compared to untreated cells in Boyden chamber assays (Figure 5C). In addition, although WNT974 suppresses migration of parental A4573 cells and scramble control cells in Boyden chamber assays, the Wnt5a knock-out clones do not respond to WNT974 (Figure 5D).

Next, we evaluated the effect of Wnt5a knock-out on other key proteins involved in Wnt-dependent cell migration, including ALCAM and vinculin. Genetic deletion of Wnt5a results in a significant decrease in vinculin expression (Figure 5A). In addition, western blotting reveals that the Wnt5a knock-out cells have a different ALCAM banding pattern (Figure 5A), suggesting a change in post-translational modification. ALCAM is a complex glycoprotein with 10 potential N-linked glycosylation sites that have been characterized in the melanoma cell line A375 as beta 1–6 branched oligosaccharides [37]. While the exact role of these N-glycans in ALCAM has not been definitively explored, the literature suggests a potential role in post-Golgi membrane sorting to the apical edge of polarized cells [38,39]. To assess the impact of Wnt5a on ALCAM glycosylation, we performed a Peptide–N-glycosidase (PNGase) assay to evaluate the presence of N-linked oligosaccharides on ALCAM. We used the amidase PNGase F which cleaves the innermost N-acetylglucosamine residue and asparagine residues of complex oligosaccharides. As shown in Figure 5E, when all cells are treated with PNGase to cleave N-linked oligosaccharides (denoted as + wells), there is not only a shift towards smaller bands in the two Wnt5a knock-out clones compared to parental and scramble control cells, but also more bands in total in the knock-out clones. These results demonstrate that the knock-out clones have more N-linked asparaginase residues that are glycosylated, suggesting that Wnt5a impacts this post-translational modification of ALCAM. Additionally, consistent with our findings in parental cells treated with WNT974, Wnt5a knock-out cells demonstrate decreased phosphorylation of FAK at tyrosine 397 at baseline in both clones in comparison to parental and scramble control cells (Figure 5F).

Confocal immunofluorescence evaluation of ALCAM and vinculin in the Wnt5a knock-out clones demonstrates an overall decrease in vinculin staining as well as more centralized clumping of ALCAM, which is similar to their localization upon treatment with WNT974 (Figure 5G). In contrast, both parental A4573 cells and scramble control cells have long protrusions containing both ALCAM and vinculin at baseline. We used Volocity to quantitate mean fluorescent intensity (MFI) in areas of co-localization of ALCAM and vinculin in all four cell lines. The amount of ALCAM and vinculin co-localization significantly increases in the Wnt5a knock-out clones compared to scramble control (Figure 5G). Notably, WNT974 treatment causes a decrease in phalloidin staining Ewing sarcoma cells (Figure 5H). In contrast, phalloidin staining is much less prominent in the Wnt5a knock-out clones at baseline and is not affected by WNT974, suggesting that the cytoskeletal structure is affected overall in these cells due to lack of Wnt5a (Figure 5H). Taken together, our findings with Wnt5a knock-out cells confirm the implications of our previous data, that Wnt5a is a key mediator of Ewing sarcoma migration through modulation of cytoskeletal structure and interactions with cell membrane-associated adhesion molecules.

## 4. Discussion

The prognosis for patients diagnosed with localized Ewing sarcoma has seen steady improvement over the past several decades, beginning with the introduction of systemic chemotherapy in the 1970s through a series of ever-more-intense multiagent chemotherapy regimens, supporting the idea that chemotherapy intensification plays a key role in improving outcomes for patients with a localized tumor. In contrast, the prognosis of patients with metastatic disease, either present from diagnosis or developed upon relapse, has not improved, suggesting that therapies targeting the pathways driving metastasis will be required to improve the survival of these patients. The past several years have, therefore, seen an increased focus on molecular mechanisms underlying metastasis. The currently accepted paradigm centers on metastasis being dependent primarily upon varying levels of expression of EWS-FLI1, with EWS-FLI1 high cells being seen as more proliferative, and EWS-FLI1 low cells having a more migratory/invasive phenotype [10,40,41,42,43]. This latter cell population is characterized by active canonical, beta-catenin-dependent Wnt signaling. Despite being implicated in the metastasis of many cancer types, there has been very little investigation of the role of non-canonical Wnt signaling pathways in Ewing sarcoma [44,45,46,47,48].

Our group previously demonstrated the importance of Wnt signaling for Ewing sarcoma metastasis by demonstrating that a pan-Wnt inhibitor, WNT974, delays the onset of metastasis in our murine model of spontaneous distant metastasis, thus prolonging survival. In that study, we found very little evidence of active beta-catenin-dependent Wnt signaling, consistent with published reports that very few Ewing sarcoma cells in patient biopsies show nuclear beta-catenin [40,49]. Our efforts to define the Wnt signaling pathway that drives metastasis revealed that a non-canonical pathway responsive to Wnt5a predominates. Supporting the concept that this non-canonical Wnt signaling pathway is important for Ewing sarcoma biology is the observation that Wnt5a is a direct transcriptional target of EWS-FLI1 [4]. Our work highlights the importance of a non-canonical Wnt5a-dependent signaling pathway in driving the cytoskeletal rearrangements and cell adhesion molecule expression changes necessary for the earliest step in the metastatic cascade migration. Notably, although we see significant effects mediated by Wnt5a, we see very little evidence of canonical Wnt signaling or of Wnt3a responsiveness, challenging the prevailing model of Wnt function in ES, which is more focused on canonical Wnt signaling.

Our data do not contradict the growing evidence supporting a role for canonical Wnt signaling in Ewing sarcoma but rather demonstrate complementary effects supporting a model integrating both canonical and non-canonical pathways. Hawkins et al. reported that canonical Wnt signaling drives changes in the tumor microenvironment that would support metastasis, such as the angiogenic switch and secretion of extracellular matrix proteins [43]. Our data integrate nicely with this model, as we show that non-canonical Wnt signaling drives an autocrine/paracrine pathway that modifies the ability of Ewing sarcoma cells to interact with the tumor microenvironment. Goodspeed et al. recently published single-cell RNA sequencing data of Ewing sarcoma tumors and found that a majority of the tumors demonstrated Wnt5a-Fzd1-mediated cell-to-cell interactions based on the expression patterns of ligands and receptors [50]. Thus, a picture emerges of canonical Wnt signaling altering cell extrinsic properties, such as the ECM and tumor vascularization, and non-canonical Wnt signaling altering cell intrinsic properties, including the actin cytoskeleton and cell adhesion molecules (Figure 6), that leverage the cell extrinsic changes driven by the canonical pathway to cooperatively drive the metastatic phenotype.

Ewing sarcoma cell lines deprived of autocrine/paracrine Wnt signals by treatment with WNT974 dramatically and specifically upregulate Wnt5a. Importantly, in our patient-derived xenograft models, Wnt5a expression is seen by immunohistochemistry in nearly all cells, in stark contrast to the very rare nuclear beta-catenin staining reported by others. Regulatory feedback loops are features that have long been studied in Wnt signaling, such as in colon cancer and glioblastoma; however, current understanding of Wnt-induced feedback is focused on beta-catenin [51,52]. Our work clearly demonstrates that there is a feedback loop operative with non-canonical, Wnt5a-dependent signaling as well in Ewing sarcoma.

Our data further demonstrates that Wnt5a drives a signaling pathway that engages multiple components identified as important in the migratory phenotypes of other cancers. In particular, we show that activation of this pathway involves phosphorylation of both FAK and Src, modification of the actin cytoskeleton, and induction of a physical interaction between the actin binding protein vinculin and the cell adhesion molecule, ALCAM. Inhibition of Wnt signaling, FAK phosphorylation, or ALCAM palmitoylation can each diminish Ewing sarcoma cell migration. In fact, deleting Wnt5a from Ewing sarcoma cells by CRISPR-mediated gene editing phenocopies treatment with the pan-Wnt inhibitor, WNT974, providing direct mechanistic evidence that Wnt5a is a key driver of a pathway that is critical for Ewing sarcoma metastasis. Changes in N-glycosylation of ALCAM in Wnt5a knock-out cells may suggest that post-translational modifications might affect Ewing sarcoma cell migration as the Wnt5a knock-out clones do not migrate well. FAK is a necessary component of both focal adhesions and invadopodia. In melanoma, when phosphorylation of FAK at tyrosine 397 is genomically blocked, cells are unable to migrate but have an increased locally invasive phenotype, suggesting that phosphorylation at this site has dual effects [53]. Phosphorylation of FAK Y397 is decreased in the majority of Ewing sarcoma cell lines with Porcn inhibition examined in this study. In Hayashi et al., the prolonged survival of the mouse ES xenografts was due solely to a delay in metastasis formation because treatment had no effect on primary tumor growth [16]. Consistent with our findings in parental cells treated with WNT974, Wnt5a knock-out cells demonstrate decreased phosphorylation of FAK at tyrosine 397 at baseline in both clones in comparison to parental and scramble control cells. The lack of impact of WNT974 on primary tumor growth while decreasing distant metastases could be due to the dual role of phosphorylation of FAK Y397 in Ewing sarcoma, similar to that of melanoma.

Src is also known to regulate actin dynamics, including directly associating with actin in colon cancer [22], and Src phosphorylation is also affected by the non-canonical signaling pathway we have defined. Phosphorylated FAK localizes Src at the leading edge of a cell, which results in polymerization of actin. When Wnt5a is knocked out, both clones A9.5 and F10.6 demonstrate statistically significant decreases in filamentous actin at baseline. These results suggest that FAK and Src signaling are not only downstream targets of Wnt5a but also similarly regulate filamentous actin formation in Ewing sarcoma. WNT974 does not alter phalloidin staining in the Wnt5a knock-out clones, providing additional support that this non-canonical Wnt ligand is the key mediator of the phenotypic effects of WNT974 on Ewing sarcoma cells.

Vinculin is a cytoskeletal protein that is a key component of focal adhesions through linkage of filamentous actin to adhesion molecules, most commonly cross-linking to membrane-bound integrins. Prior research has demonstrated that vinculin-null mice embryo fibroblasts are viable but only able to generate small non-functional focal adhesions and demonstrate significant defects in organization of the actin cytoskeleton and cell spreading [54]. In addition, vinculin protein levels have been associated with progression in prostate cancer and NSCLC [55,56]. Our data demonstrates that vinculin physically associates with ALCAM, an association that increases with PORCN inhibition, which decreases migratory activity. These results support a model that increased association between these two proteins negatively impacts the cell’s ability to turn over focal adhesions, a necessary step for cell migration. Additionally, vinculin protein levels are decreased in the Wnt5a CRISPR clones which demonstrate decreased filamentous actin, suggesting that loss of Wnt5a which leads to decreased vinculin may directly result in a lack of filamentous actin and cell spreading, similar to what has been shown in vinculin-null fibroblasts.

The study of the immunoglobulin family of cell adhesion molecules has been an evolving area of interest in cancer because both the presence of these molecules and their distribution throughout the tumors have been shown to be clinically impactful, such as L1CAM in endometrial and colon cancers [57,58]. There is currently an open phase I clinical trial evaluating autologous lentivirally induced L1CAM T cells in prostate cancer [59]. Wnt5a expression is not only correlated with a more invasive phenotype in ER+ breast cancer, but is also associated with increased ALCAM expression, leading Kobayashi et al. to hypothesize that Wnt5a increases the invasiveness of breast carcinoma by inducing ALCAM expression [60]. Yang et al. have reported a role for ALCAM in Ewing sarcoma cell migration mediated through small Rho GTPases [38]. Their study demonstrates signaling between numerous important targets of cell migration that have previously been identified in genetic screens in Ewing sarcoma, but not reported as a pathway [20,21,61,62,63].

Our data demonstrate a significant role of Wnt5a in the formation of focal adhesions in Ewing sarcoma. It is known that FAK controls the balance between invadopodia and focal adhesion formation by regulating the localization of Src [64]. This role is important in the context of our data because focal adhesions are primarily involved in cell migration and attachment to extracellular matrix whereas invadopodia are invasive adhesions that can degrade the extracellular matrix. Our data demonstrate that a lack of Wnt5a in Ewing sarcoma cells affects cell migration and the phenotypic changes of the Wnt5a knock-out clones as well as the fact that they require a matrix of fibronectin to adhere well is consistent with the function of focal adhesions. In contrast, our data did not show changes in invasion (Appendix A). Mechanistically, these findings suggest that a population of Wnt5a high-expressing cells in the primary tumor likely drive focal adhesion formation to initiate cell migration and ultimately attachment at a distant location, contributing to metastases. Our previous xenograft work demonstrated a statistically significant decrease in time to metastases; therefore, targeting this novel pathway could be a clinically relevant adjuvant therapy to cytotoxic agents to inhibit metastatic development. Thus, our work strongly supports a more nuanced model of Ewing sarcoma metastasis, involving both canonical and non-canonical Wnt signaling pathways which can modify both the ECM and the way cells respond to their environment, and would also support the development of clinical trials focused on disrupting these pathways to prevent metastatic relapses and improve patient outcomes.

## 5. Conclusions

Ewing sarcoma is the second most common bone malignancy in children, adolescents, and young adults. Despite improvements in overall survival for patients with localized disease, those with metastatic disease continue to face poor outcomes with an approximate 25% 5-year overall survival. In addition, approximately a third of patients who initially present with localized disease will ultimately relapse with metastatic disease. Therefore, a better understanding of drivers of cell migration and metastasis is necessary in Ewing sarcoma. Here, we present a novel intrinsic Wnt5a-responsive pathway that drives formation of focal adhesions through phosphorylation of FAK and Src, leading to cross-linking of actin filaments via vinculin with resultant anchorage to the cell membrane through ALCAM. Ewing sarcoma cells appear to be heterogeneous with prior studies demonstrating different levels of EWS;FLi1 transcripts in different states of cell cycle and sparse activation of nuclear beta-catenin in clinical samples, so our work highlights the importance of understanding non-canonical Wnt signaling in the context of the earliest steps in the metastatic cascade migration. To support further development of clinical trials for Ewing sarcoma, we conclude that there is a more nuanced model of Ewing sarcoma metastasis which involves both canonical and non-canonical Wnt signaling.

## Figures and Tables

**Figure 1 cancers-17-03712-f001:**
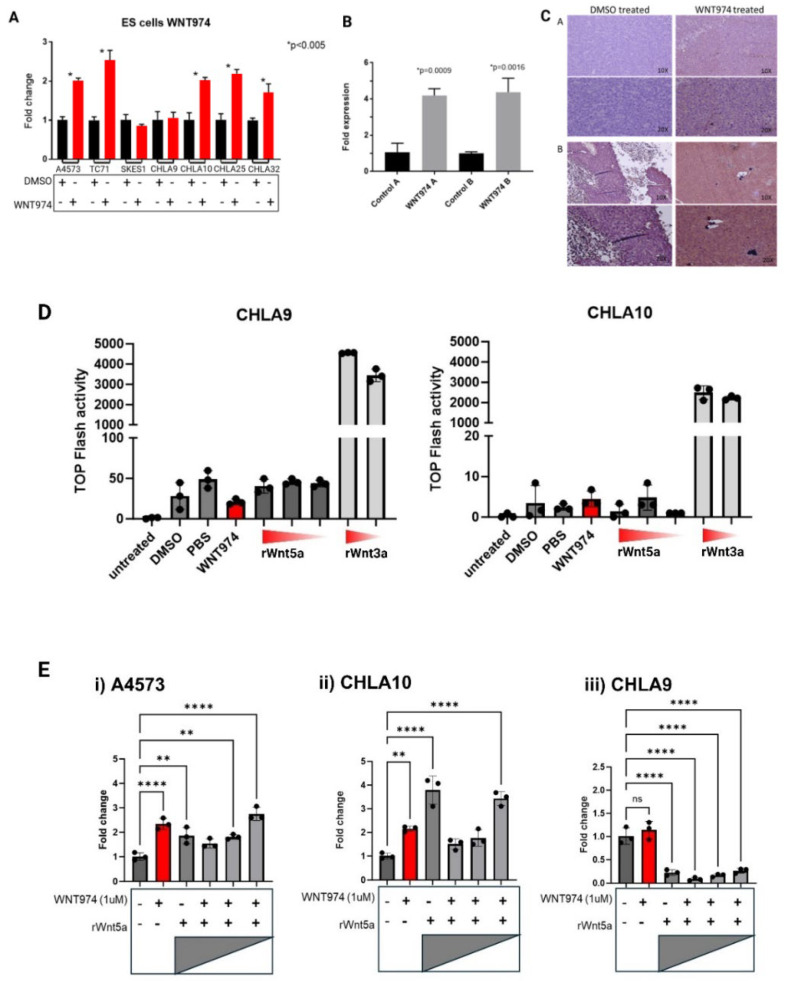
Expression of a non-canonical Wnt ligand, Wnt5a, is modulated by WNT974 in Ewing sarcoma cells. (**A**) Ewing sarcoma cell lines were treated with or without WNT974, and expression of Wnt5a was determined by quantitative RT-PCR. Error bars represent standard error of the mean of triplicate experiments, and * indicates a *p* value < 0.05. (**B**) Wnt5a mRNA levels were evaluated by quantitative RT-PCR in Ewing sarcoma PDX tumors growing in NSG mice treated with either DMSO (control) or WNT974. Error bars represent standard error of the mean of triplicate experiments, and *p* values are indicated. (**C**) Wnt5a protein expression was evaluated by immunohistochemistry in Ewing sarcoma PDX tumors from NSG mice treated with DMSO or WNT974. Panels (A and B) are from individual tumors. Magnification is indicated in each frame. (**D**) TOPFlash luciferase assay demonstrates high activity only in the presence of recombinant Wnt3a (1 μg/mL and 250 ng/mL), and little to no activity in the presence of Wnt5a (1 μg/mL, 500 ng/mL, and 250 ng/mL, respectively). (**E**) Wnt5a mRNA expression was evaluated by quantitative RT-PCR in Ewing sarcoma cell lines treated with or without WNT974, as well as increasing concentrations of exogenous recombinant Wnt5a protein (1 μg/mL, 500 ng/mL, and 250 ng/mL, respectively). A4573 (i) and CHLA10 (ii) cells show a distinct response pattern compared with the nonmetastatic CHLA9 cell line (iii). Each experiment was repeated a minimum of three times. Error bars represent standard error of the mean of triplicate experiments, and asterisks indicate the degree of statistical difference between indicated conditions * *p* </= 0.05, ** *p* </= 0.01, and **** *p* </= 0.0001.

**Figure 2 cancers-17-03712-f002:**
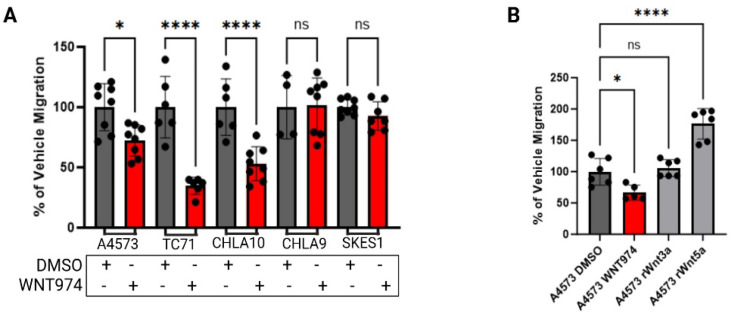
Ewing sarcoma cell migration is regulated by Wnt5a. (**A**) WNT974 decreased the migration of Wnt5a-responsive Ewing sarcoma cell lines A4573, TC71, and CHLA10, but not the migration of the unresponsive cell lines CHLA9 and SK-ES-1, towards FBS in Boyden chamber assays. (**B**) Recombinant Wnt5a, but not recombinant Wnt3a, is a strong chemoattractant to A4573 cells in Boyden chamber assays. (**C**) Inclusion of rWnt5a in the upper chamber inhibited migration of Ewing sarcoma cells toward FBS (group A), whereas including rWnt5a in the lower chamber, along with FBS, augmented migration (group B). This effect is abolished by the inclusion of WNT974 in the upper chambers (group C). In each experiment, error bars indicate standard error of the mean, and asterisks indicate the degree of statistical significance. Each experiment was repeated a minimum of three times. * *p* </= 0.05, ** *p* </= 0.01, *** *p* </= 0.001, **** *p* </= 0.0001.

**Figure 3 cancers-17-03712-f003:**
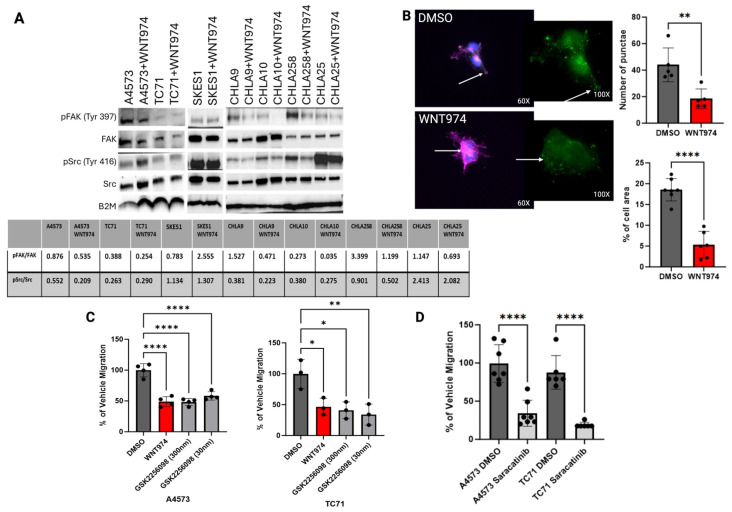
(**A**) Western blot for pFAK (Tyr 397) and pSrc (Tyr 416) in seven Ewing sarcoma cell lines treated with vehicle control (DMSO) versus WNT974, with a decrease in both pFAK and pSrc in the majority of cell lines treated with WNT974. (**B**) Immunofluorescence example for pFAK (Tyr 397) in A4573 cells. Note that pFAK (green) is found in the protrusions of the DMSO-treated A4573 cells but is found in the centralized clumps of actin (pink) in the WNT974-treated cells. With WNT974 treatment, there is a statistically significant decrease in pFAK puncta and the amount of pFAK throughout the cells. Green-fluorescent insert demonstrates pFAK with dots representative of Volocity analysis. (**C**) Boyden chamber assay using two different doses of FAK inhibitor GSK2256098 in A4573 and TC71 cells. Note that migration is significantly impaired in both cell lines upon treatment with the FAK inhibitor, even at nanomolar doses. (**D**) Treatment with Src inhibitor, saracatinib (1 μM), also results in a statistically significant decrease in cell migration, also suggesting that Src is important for Ewing sarcoma cell migration. In addition to catalytic proteins, focal adhesions contain adaptor proteins that mediate interactions between the cytoskeleton and the extracellular matrix [31]. ALCAM is an adaptor protein that contains immunoglobulin-rich domains with the amino-terminal V-type immunoglobulin domain required for cell-to-cell adhesive interactions. ALCAM can form homotypic or heterotypic interactions, and its expression is highest at areas of cell-to-cell contact where it can interact with other cell adhesion molecules [23,24]. ALCAM is expressed in over 70% of pediatric sarcomas, so we started by assessing changes in protein expression with WNT974 treatment [32]. ALCAM protein levels were unchanged with WNT974 treatment, so we next evaluated ALCAM distribution in the cell (Supplementary Appendix AE). As shown in Figure 4A, the vehicle control-treated cells maintain a small number, typically 2–3, of well-formed long protrusions in which ALCAM can be found throughout, in addition to its presence in the perinuclear region of the cytoplasm. In contrast, in cells treated with WNT974, there is bright centralized clumping of ALCAM and a notable loss of long protrusions in most of the cells. The highest concentration of ALCAM in these cells appears to overlap the nucleus, in contrast to the perinuclear cytoplasmic distribution in the vehicle control-treated cells. In addition, many of the WNT974-treated cells demonstrate a more circumferential ruffled appearance to the plasma membrane. In addition, there is a decrease in the number of long protrusions from the cells with WNT974 treatment (Figure 4A). ALCAM is a member of a family of cell adhesion molecules. Another family member that has been implicated in sarcoma biology is MCAM [19,33,34]. The involvement of ALCAM in Wnt5a-mediated cytoskeletal rearrangements is specific, because no such changes were appreciated with MCAM immunofluorescence (Figure 4A). * *p *</= 0.05, ** *p *</= 0.01, **** *p *</= 0.0001.

**Figure 4 cancers-17-03712-f004:**
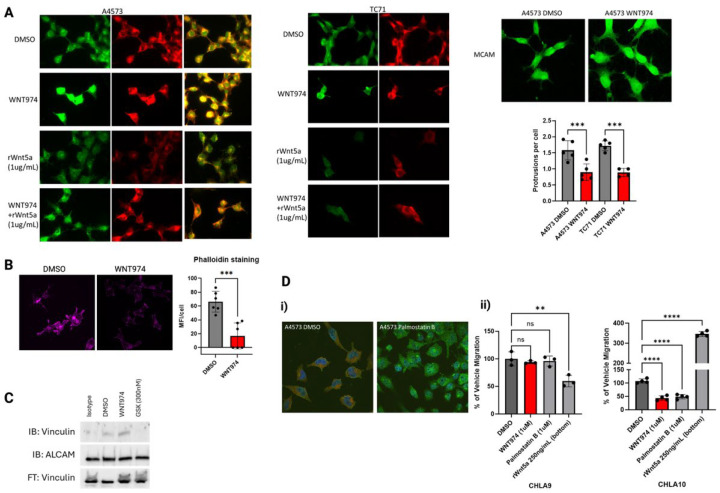
(**A**) Immunofluorescence of ALCAM (green) and vinculin (red) in A4573 cells and TC71 cells. Note the co-localization of both proteins and the centralized clumping with WNT974 treatment in both cell lines. When given recombinant Wnt5a, ALCAM and vinculin both appear less bright versus if the cells are treated with WNT974 and then given that same dose of recombinant Wnt5a. This suggests that the cells respond to the amount of Wnt5a in the environment which affects ALCAM and vinculin. Additional panel demonstrating similar findings in TC71 cells. MCAM immunofluorescence does not demonstrate a redistribution pattern upon WNT974 treatment, in contrast to ALCAM. Quantification of protrusions per cell demonstrating a decrease in protrusions upon WNT974 treatment. (**B**) Phalloidin staining of filamentous actin is statistically significantly decreased in A4573 cells treated with WNT974 compared to vehicle control. (**C**) Immunoprecipitation of vinculin and ALCAM in A4573 cells demonstrating a significant increase in association between the two proteins upon WNT974 treatment. Notably, there is a complete dissociation between the proteins upon FAK inhibition treatment. Both changes are different from vehicle control-treated cells, suggesting that changes to how much these two proteins are in contact affects cell migration. (**D**) (**i**) Using palmostatin B, an inhibitor of acyl-protein thioesterase 1 which depalmitoylates cell surface receptors to allow for lateral migration through the membrane, A4573 cells demonstrate a centralized clumping of ALCAM. (**ii**) Additionally, WNT974 and palmostatin B only inhibit migration in Boyden chamber assays in CLHA10, the metastatic-derived cell line, but not in CHLA9, the localized-derived cell line. Only CHLA10 responds to the lowest amount of recombinant Wnt5a as a chemoattractant by significantly increased cell migration to approximately 400% of vehicle migration, consistent with the hypothesis that Wnt5a may be more active in the metastatic process. Each experiment was repeated a minimum of three times. Error bars represent standard error of the mean of triplicate experiments, and asterisks indicate the degree of statistical difference between indicated conditions ** *p* </= 0.01, *** *p* </= 0.001, and **** *p* </= 0.0001.

**Figure 5 cancers-17-03712-f005:**
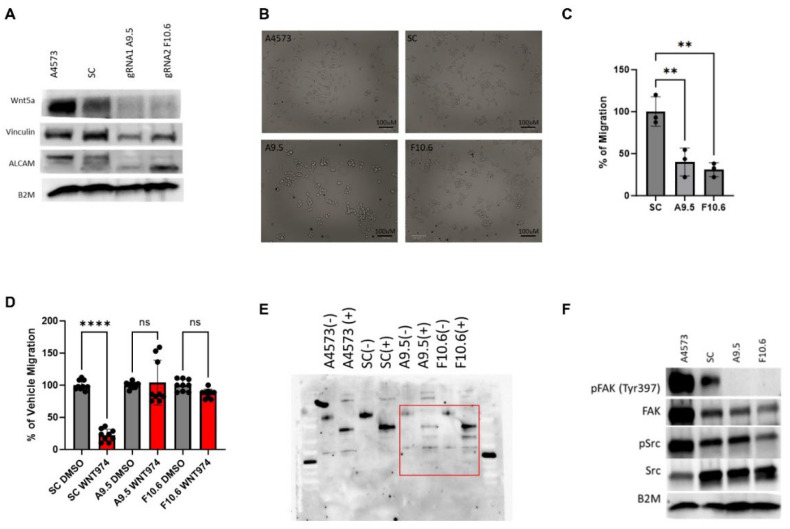
CRISPR-Cas9 gene editing confirms the critical role of Wnt5a in Ewing sarcoma migration. (**A**) Western blotting confirms that the Wnt5a CRISPR-Cas9 clones A9.5 and F10.6 lack Wnt5a protein expression and show a decrease in both vinculin and ALCAM expression as well as a shift in ALCAM banding patterns compared with the parental cells A4573. Beta-2-microglobulin (B2M) is a loading control. (**B**) Photomicrograph of Wnt5a CRISPR-Cas9 clone A9.5 which grows as clumps of cells with bright GFP+ signal (100 μm). (**C**) In Boyden chamber assays, the Wnt5a CRISPR-Cas9 clones migrate statistically significantly less than the scramble control. (**D**) In Boyden chamber assays using FBS as a chemoattractant, the Wnt5a CRISPR-Cas9 clones demonstrate impaired migration in comparison to scramble control. In panels C and D, error bars represent standard error of the mean, and asterisks reflect the degree of statistical significance. Experiments were repeated three times. (**E**) PNGase assay of ALCAM in parental A4573 cells, scramble control, and the two Wnt5a knock-out clones demonstrate a different banding pattern in the Wnt5a knock-out clones when treated with PNGase (denoted as +) compared with parental cells or scramble control. (**F**) Western blotting demonstrates a lack of phospho-FAK in the Wnt5a knock-out clones. (**G**) Immunofluorescence analysis of parental A4573 cells, scramble control cells, and the two Wnt5a knock-out clones shows that the parental cells have an average of 2–3 large protrusions and co-localization of ALCAM (green) and vinculin (red) upon treatment with vehicle control (DMSO) whereas upon WNT974 treatment, the cells appear to have decreased vinculin as well as centralized perinuclear clumping of ALCAM. The same pattern is also seen in the scramble control. In contrast, both the morphology of the Wnt5a knock-out clones A9.5 and F10.6, as well as the distribution of ALCAM and vinculin, is not affected by WNT974. The Wnt5a knock-out clones, A9.5 and F10.6, have a statistically significantly increased colocalization between ALCAM and vinculin when treated with WNT974 (quantified by MFI in the fused channel), a contrast to the decreased co-localization seen in scramble control cells treated with WNT974. To the right is quantification of co-localization. Error bars represent standard deviation. (**H**) Staining of cells with fluorescently tagged phalloidin shows significantly less filamentous actin in Wnt5a knock-out clones compared with parental cells and scramble control (60X). Treatment of parental cells and scramble control with WNT974 decreases phalloidin staining, an effect not seen in the Wnt5a knock-out clones. To the right is quantification of the phalloidin staining. Error bars represent standard deviation, and asterisks indicate the degree of statistical significance. Each experiment was repeated a minimum of three times. * *p* </= 0.05, ** *p* </= 0.01, **** *p* </= 0.0001.

**Figure 6 cancers-17-03712-f006:**
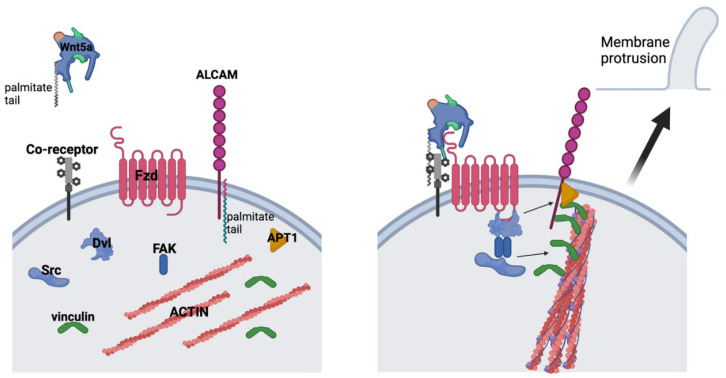
Working model of the role of Wnt5a in Ewing sarcoma cells. Upon Wnt5a binding with its Fzd receptor(s) and potential co-receptor(s), FAK (Tyr 397) becomes phosphorylated and dimerizes to phosphorylate Src (Tyr 416). Phosphorylated Src then phosphorylates vinculin, allowing it to bind to F-actin bundles and cross-linking it to ALCAM. ALCAM is depalmitolyated by APT1 to allow it to move within the cell membrane, thereby deforming the membrane at the site of proximity to actin bundles.

## Data Availability

The original contributions presented in this study are included in the article/Appendix A. Further inquiries can be directed to the corresponding authors.

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
