# Peer review of "Wnt5a Regulates Focal Adhesion Formation to Promote Migration in Ewing Sarcoma"

_cancers, 2025, doi:10.3390/cancers17223712_

Round 1

Reviewer 1 Report

Comments and Suggestions for Authors

Several items in the manuscript were not clear and need further clarification and explanation: 

  1. The introduction is rather deficient. Many points in the results can be moved into the introduction, particularly the information on ALCAM, FAK, Src and other adhesion molecules in pages 10,11. What is the rationale for studying these molecules in relation to WNT signaling and WNT5a?
  2. Specify the number of cell lines in line 99. Also include the EWS fusion or rearrangement status of the studied cell lines
  3. More detailed method description is needed on CRISPR editing of Wnt5a clones to generate wnt5a-knock out cell lines, preferably in a separate subheading
  4. Judging from Fig 1B and 1C, it appears that NSG mice were used in the experiments. However, xenograph animal experiments are not described. These need to be included in its own subheading in the method section
  5. It would make the results easier to follow if subheadings were inserted in pages, 7, 9, 11 and 12
  6. Figure 3 is too crowded with many subpanels. Suggest breaking it into 2-3 figures 
  7. While experiments described the effect of WNT974 on cell lines migration, its effects on xenograph animal models were not adequately described. To confirm concordance with in vitro studies, the authors should study and present the expression of pFAK, pSRC, ALCAM, vinculin and other molecules in animal models localized and metastatic tumors with and without WNT974 treatment.  
  8. It is not clear from the discussion how WNT974 can directly affect Ewing's sarcoma viability and whether this can be used as a potential treatment modality. While cell migration is decreased by WNT974 (lines 320-321), the authors discussed lack of effect on tumor growth (line 667). An ideal therapeutic target would inhibit both cell proliferation and migration. 

Reviewer 2 Report

Comments and Suggestions for Authors

In this paper, Baker et al. investigate the role of Wnt5a on the metastatic potential of EWS. The authors elegantly show that non-canonical Wnt signalling drives metastasis in EWS, primarily via Wnt5a, which works by facilitating cell migration via altered adhesion mechanisms. This is an important study as survival for EWS has not improved in recent decades, and current protocols fail to meet the challenges of metastatic disease. 

This is a high-quality study that lends an important piece of information to the field that can be potentially translated into clinical practice. The manuscript is clear and well-written, with sufficient background and relevant references.

I have no suggestions for improvements. 

Author Response

We are grateful that the reviewer believe the manuscript is acceptable without modifications.

Round 2

Reviewer 1 Report

Comments and Suggestions for Authors

None

Author Response

We have amended the Methods section as requested